# Improving Intent Classification Using Unlabeled Data from Large Corpora

Gabriel Bercaru [1,2,*] , Ciprian-Octavian Truică [1,2,*] , Costin-Gabriel Chiru [1,2,*] and Traian Rebedea [2,*]

1 SoftTehnica, RO-030128 Bucharest, Romania
2 Computer Science and Engineering Department, Faculty of Automatic Control and Computers, University Politehnica of Bucharest, RO-060042 Bucharest, Romania
* Correspondence: gabriel.bercaru@upb.ro (G.B.); ciprian.truica@upb.ro (C.-O.T.); costin.chiru@upb.ro (C.-G.C.); traian.rebedea@upb.ro (T.R.)

**Abstract:** Intent classification is a central component of a Natural Language Understanding (NLU) pipeline for conversational agents. The quality of such a component depends on the quality of the training data, however, for many conversational scenarios, the data might be scarce; in these scenarios, data augmentation techniques are used. Having general data augmentation methods that can generalize to many datasets is highly desirable. The work presented in this paper is centered around two main components. First, we explore the influence of various feature vectors on the task of intent classification using RASA's text classification capabilities. The second part of this work consists of a generic method for efficiently augmenting textual corpora using large datasets of unlabeled data. The proposed method is able to efficiently mine for examples similar to the ones that are already present in standard, natural language corpora. The experimental results show that using our corpus augmentation methods enables an increase in text classification accuracy in few-shot settings. Particularly, the gains in accuracy raise up to 16% when the number of labeled examples is very low (e.g., two examples). We believe that our method is important for any Natural Language Processing (NLP) or NLU task in which labeled training data are scarce or expensive to obtain. Lastly, we give some insights into future work, which aims at combining our proposed method with a semi-supervised learning approach.

**Keywords:** intent classification; chatbot; few-shot learning; data augmentation; online clustering; data projection

**MSC:** 68T50

## 1. Introduction

In the present day, conversational agents (or chatbots) are a core component of many applications, ranging from online reservations to customer support. The quality of the chatbot replies depends on its ability to accurately understand the user query. To ensure that, the user intention must be understood. Thus, the chatbot designer has to provide multiple alternatives on how the user may formulate a query. This is often performed by specialists and it is a time-consuming task. Automating this process will highly benefit chatbot designers, reducing iteration time.

In order to obtain chatbot training examples in a semi or fully automated manner, one could leverage large volumes of unlabeled data available in various online corpora (e.g., movie subtitles, translation datasets, etc.). The only impediment is that the unlabeled corpora are usually large enough such that a simple search for semantically similar examples within them becomes unpractical. As a result, efficient retrieval of similar examples is highly desirable.

Driven by such motivation, we propose a novel pipeline for efficiently analyzing large, unlabeled corpora and extracting examples similar to a user-supplied query. We aim to

minimize the retrieval time while maintaining a high similarity between the query and the retrieved example. Moreover, we examine how the proposed example retrieval system improves the intent classification accuracy in several few-shot learning scenarios, where intent examples are scarce.

As the results of this research will show, our proposed method is highly beneficial in the few-shot intent classification scenario. In such a setup, the number of labeled examples is very small (e.g., 2, 3, 5, or 10 examples per class). Using our similar example retrieval pipeline, we expand the number of examples per class, while increasing the classification accuracy with significant rates, up to 16%. To our knowledge, at the time of writing this article, there is no open-source service that can be used to augment textual datasets based on online clustering of movie conversations. Our method allows relatively quick and cheap dataset augmentation, making use of only open-source components.

The research questions that guide our research can be summarized as follows:

$(Q_1)$ How do existing chatbots perform in terms of intent classification?
$(Q_2)$ How can we use unlabeled data to improve the intent detection phase of conversational agents?
$(Q_3)$ How can we efficiently extract meaningful examples from large, unlabeled corpora?
$(Q_4)$ To what extent does the proposed system benefit in a few-shot learning scenario?

Our objectives can be stated as follows:

$(O_1)$ Analyze current intent classification performance for existing systems to address $Q_1$;
$(O_2)$ Process large, unlabeled corpora such that they become suitable for our similarity-based example retrieval system to address $Q_2$;
$(O_3)$ Achieve low example retrieval duration to address $Q_3$;
$(O_4)$ Evaluate our example retrieval system in few-shot learning scenarios to address $Q_4$.

The contributions of this work can be summarized as follows:

$(C_1)$ An analysis of standard intent detection systems and their performance;
$(C_2)$ An efficient, similarity-based retrieval system that is used for augmenting intent classification datasets;
$(C_3)$ An extensive experimental performance analysis of our proposed system in few-shot learning scenarios using real-world datasets.

The rest of this paper is structured as follows. Section 2 presents previous work done with respect to intent classification, in both standard and few-shot scenarios. Section 3 shows the general structure of a RASA-based conversational agent. Section 4 describes the experiments performed on the RASA NLU component, for analyzing Transformer models' accuracy in intent classification. Section 5 provides details on the work done for retrieving similar examples in a large corpus. Section 6 presents the evaluation methods and the results obtained for corpus clustering. Section 7 discusses our findings and their implications. Lastly, Section 8 summarizes and concludes our work and hints at possible future directions.

## 2. Related Work

The first component of our contribution consists of an analysis of existing Transformer embedders in the context of intent classification. Balakrishnan et al. [1] provided a similar analysis for disaster classification in tweets. As their results show, using Transformer-based embedders is beneficial and increases the accuracy, compared to other embedding options, e.g., bag-of-words, Word2Vec, GloVe, etc.

As already mentioned, one of the crucial aspects regarding the quality of a chatbot is related to the datasets that it uses. Related to this, Larson et al. [2] and Casanueva et al. [3] introduced two datasets for intent classification, namely CLINC150 and BANKING77. CLINC150 is designed for benchmarking models meant to distinguish in-domain queries from out-of-domain queries, thus its structure is more complex compared to BANKING77. CLINC150 queries span 150 intents over 10 different domains, while BANKING77 queries

span 77 intents over a single domain, namely banking-related operations. In this paper, we use these two datasets to evaluate the performance of our models.

Another important aspect of a chatbot is the framework used for its development. In this sense, Liu et al. [4] presented an analysis of several conversational agents designing frameworks, including RASA. Their study focuses on a dataset created by the authors, which includes queries belonging to 21 domains, with 64 intents and 54 annotated entity types. The queries belonging to the mentioned dataset contain tasks that can be given to a house-cleaning robot. Compared to this dataset, the ones used in this paper contain more intents, namely 77 for BANKING77 and 150 for CLINC150.

Most of the current literature is centered around two axes: intent identification and data augmentation. Regarding intent identification, Ahmadvand et al. [5] performed dialogue act classification in the context of open-domain conversational agents. Unlike our subject, open-domain dialogue cannot divide the intents into well-defined classes simply by looking at the current utterance. Consequently, the authors tackled the problem by incorporating dialogue history information. The information is encoded by including features from the lexical, syntactic, and system state information layers. The information is captured through pre-trained Word2Vec embedding vectors. The training procedure is split across two distinct phases: (1) the dialogue act system is trained on human-to-human conversations, and (2) the human-to-machine conversations are fine-tuned. Their results, evaluated on Switchboard data and Alexa Prize data, show that the proposed Context-aware Dialogue Act Classification system outperforms state-of-the-art models trained on each dataset.

Zhan et al. [6] designed an out-of-scope (OOS) intent detection method, modeling the distribution of out-of-scope intents. Their work splits OOS intents into (1) 'hard' OOS intents that are close to the decision boundary, and (2) 'easy' intents that are distant from the in-scope intents. Their research is focused on a rather binary classification task, namely separating in-scope from out-of-scope intents. Nonetheless, the datasets used for carrying out the research include BANKING77 and CLINC150, the same as our work does. The authors tested their models by using only 25%, 50% or 75% of the classes (in three different setups), while leaving the rest of the classes unseen. The models are subsequently used to predict whether an example is in-scope or out-of-scope. The best results are obtained in the 75% seen–25% unseen classes setup, with 88.08% accuracy for CLINC150 and 81.07% accuracy for BANKING77.

In intent classification, out-of-scope intents can be further divided into two classes [7]: (1) in-distribution out-of-scope examples (ID-OOS), and (2) out-of-distribution out-of-scope examples (OOD-OOS). Zhang et al. [7] showed that pre-trained Transformer models (e.g., BERT, RoBERTa, etc.) are vulnerable to mispredicting OOD-OOS examples. However, existing intent classification datasets, such as CLINC150 and BANKING77 do not contain any ID-OOS data. Particularly, CLINC150 contains an OOS class, but most of the examples are easily distinguishable from the in-domain ones, thus OOD. Besides the performance analysis of pre-trained Transformer based models on the OOD-OOS examples, the authors contributed with two datasets for OOS intent detection. These datasets feature both ID-OOS and OOD-OOS data.

Liu et al. [8] tackled the problem of intent classification when the number of available examples per intent is limited. They reconstructed capsule network models (such as IntentCapsNet [9]) in order to include information regarding the possible polysemy of the words which contribute to the features of the semantic capsules. Moreover, their proposed method, IntentCapsNet-ZS, behaves better than previous models with respect to unseen intents, in the zero-shot setting.

Yan et al. [10] designed a Gaussian Mixture Model (GMM) method for out-of-domain intent detection. Their research shows that previous intent outlier detection methods project sentence embeddings into a latent space in which the class (intent) label is the centroid and all examples are scattered across a long and narrow domain. In such representation, detecting out-of-scope intents is error-prone. Their proposed method alleviates this problem

by regularizing the projection space such that the class label remains the centroid, but the examples are distributed more evenly around it. The output of such a scenario can be paired with an anomaly detection algorithm in order to separate in-domain intents from unknown out-of-domain intents. Moreover, the authors demonstrated that their method (SEG—Semantic-Enhanced Gaussian Mixture Model) can be paired with previously developed zero-shot intent classification methods (i.e., ReCapsNet [11]), in order to improve their performance.

In terms of data augmentation, Chatterjee and Sengupta [12] performed a corpus clustering operation, with the goal of grouping together similar sentences in a corpus, for manual intent annotation. With their technique, the resulting corpus may be used for manually augmenting the dataset of any intent classification task. Their intent discovery pipeline comprises 4 main steps: (1) the extraction of conversation utterances using a pre-trained dialogue act classifier, (2) grouping together similar utterances, (3) manual labeling of the clusters, and (4) re-classifying utterances that have not been previously assigned to any cluster. The experimental results show that a clustering algorithm such as ITER-DBSCAN performs better than previous methods when it comes to intent coverage. Unlike their work, our proposed corpus augmentation method does not require any manual intervention of the designer of the conversational agent. Similarly, Kuchlous and Kadaba [13] performed intent classification in the context of a therapy and mental wellness-oriented chatbot. Their dataset is of rather limited size, containing only 4 classes (intents), with approximately 400 examples in total. The authors used this dataset to benchmark several non-neural based models: Multinomial Naïve Bayes, Logistic Regression, SVM, and Random Forest. Due to the limited dataset, the authors resorted to several processing steps, i.e., artificially augmenting the training set and building a custom English stop words list. By applying these steps, the accuracy of the classification is increased. Unlike their work, in our experiments, we use a standard English language stop words list.

Sahu et al. [14] designed a method of augmenting datasets for intent classification that employs large language models (such as GPT-3 [15]) for generating artificial training examples, given a context containing the original intents for a specific class. However, their method requires the execution of two expensive stages in the pipeline: (1) using a large language model for performing inference on all the available examples, and (2) the possibility of including a manual verification stage, in order to filter out unrelated, retrieved examples. Furthermore, the authors investigated the effect of their corpus augmentation method in few-shot learning scenarios. Compared to their method, our experiments do not require large language models to augment the corpus and the post-processing filtering is performed automatically.

## 3. RASA Components for Building Conversational Agents

RASA [16] is one of the most successful frameworks for building conversational agents. Its architecture is composed of several interconnected modules, which can function both independently or as a whole.

A powerful feature of RASA is the possibility to integrate state-of-the-art, pre-trained Transformer models, via the Huggingface library [17] (https://huggingface.co/ (last accessed on 7 November 2022)). These Transformers can increase the intent prediction accuracy, in the NLU phase, by providing their own embedding vectors for the supplied tokens, at the cost of a larger memory footprint.

In the context of task-oriented dialogue, RASA emerged as the preferred solution, due to its ability to handle both simple (query-answer) and complex (multiple turns needed to obtain the required information) conversational scenarios. Its structure is composed of two loosely coupled sub-systems: the natural language understanding (NLU) component and the dialogue management component.

The NLU component is responsible for extracting information at a single dialogue turn, e.g., the intent associated with the turn and possible entities in the sentence. This process is divided throughout a pipeline consisting of several stages: (1) a tokenizer (which

splits the raw input text into tokens), (2) one or several featurizers (which encode and extract meaningful information from the tokens), and (3) a classification method, which produces the final intent and entities associated with the input sentence. One of the more important stages in the pipeline is the featurizer stage where multiple methods of encoding the tokens are available. The encoding mechanism can employ either 'standard' text-based metrics (TF-IDF scores) or embedding vectors obtained through neural models (e.g., word embeddings, Transformer-based models, etc.).

Furthermore, the dialogue management component dictates how the conversation evolves. This component utilizes three main policies which choose the next agent action, given the dialogue context:

(1) *Rule policy.* If the current user input matches one of the agent's known rules, the corresponding action is executed immediately, without taking into consideration the conversation history or the known scenarios.

(2) *Memorization policy.* Unless the current turn matches any rule, the agent tries to fit it inside one of the conversational scenarios. A scenario consists of several exchanges between the user and the agent.

(3) *TED policy (Transformer Embedding Dialogue policy)* [18]. When the input text does not match any of the predefined rules or scenarios, the agent attempts to choose the most probable of the known actions, given the context. This is achieved by (i) generating the embedding of the input text using a Transformer encoder, (ii) computing the similarity between the resulting embedding vector and known actions embeddings, and (iii) extracting any possible entities in the user text through a Conditional Random Field (CRF) layer.

## 4. RASA NLU Intent Classification

In RASA, the NLU and dialogue management components are loosely coupled—the RASA NLU component can function independently of the latter one. As a result, the intent classification experiments are conducted using only the NLU stage.

### 4.1. Datasets Used

There are many public datasets (https://github.com/clinc/nlu-datasets (last accessed on 7 November 2022)) available online for benchmarking the intent classification task. For our experiments, we use both CLINC150 and BANKING77 datasets.

CLINC150 [2] is a dataset proposed for evaluating the performance of out-of-scope classification systems. The main version of the dataset (*full*) contains 150 in-domain classes and one class for out-of-domain examples. Each of the 150 domain classes contains 100 training examples, 20 validation examples, and 30 test examples. The out-of-domain class is split into 100 training examples, 100 validation examples, and 1000 test examples.

Besides the *full* dataset, Larson et al. [2] proposed 3 more datasets as sub-samples of the original large one. The *small* version of CLINC150 follows the same class distribution. However, it contains fewer examples for training, i.e., 50 examples per class. The *imbalanced* version of the dataset poses additional challenges since training examples are no longer equally distributed across classes. Thus, intents have either 25, 50, 75, or 100 training examples. The *plus* version features more training examples per class, i.e, 250.

BANKING77 [3] is another dataset introduced for benchmarking text classification methods. However, this dataset contains only queries from the banking domain. These banking queries are divided across 77 in-scope classes. It is a balanced set, as all intents contain the same number of examples.

### 4.2. Intent Classification

Within the RASA framework, accurately classifying the intent encoded inside a user query is critical for a correct dialogue flow. Consequently, RASA provides numerous options for analyzing the input text and extracting meaningful features, which ultimately determine the intent.

While most of the RASA pipeline components are customizable, the used featurizers deserve more attention as choosing one type of featurizer may have implications beyond classification accuracy. The memory footprint of the featurizer and the overall response time of the system are also metrics to consider.

One of the simpler featurizers tested is the *CountVectorsFeaturizer* (https://rasa.com/docs/rasa/components/#countvectorsfeaturizer (last accessed on 7 November 2022)), which analyzes the user text and creates a bag-of-words representation based on it. The result is a sparse representation of the input sequence, which disregards token sequentiality. In the case of a task such as intent classification, sequentiality might not prove to be as important as for other NLP/NLU tasks (i.e., machine translation, named entity recognition, part of speech tagging), as in many cases the intent of a sentence is determined by a keyword irrespective to the position it is located. Sparse tokenizers are able to extract features at multiple n-gram granularity (standard *n* values range between 1 and 4), working either at word or character level. For evaluation, we featurize the text based on character n-grams with sizes between 2 and 4 characters. Note that n-gram extraction is performed on each word's lemma rather than on the original word.

In order to better capture semantic similarities between words, several types of dense featurizers can be used, e.g., featurizers that produce embedding vectors based on the user utterance. We test the following dense featurizers: (1) *SpacyFeaturizer*, and (2) multiple *LanguageModelFeaturizers*. In the case of *SpacyFeaturizer*, the intent classifier used is *SklearnIntentClassifier* (implemented through Scikit-learn [19]). *SklearnIntentClassifier* is based on a linear SVM classification algorithm for which the parameters are determined via GridSearchCV. The *LanguageModelFeaturizers* component allows embedding integration mechanisms from state-of-the-art Transformer-based language models. To this extent, our experiments employ 6 models: (1) BERT [20], (2) ConveRT [21], a Transformer-based encoder designed for conversations, (3) RoBERTa [22], (4) GPT [23], (5) GPT-2 [24], and (6) XLNet [25]. For all the language model featurizers, the extracted features are used as input for *DIETClassifier* [18], a multi-task model for intent classification and entity extraction. DIETClassifier uses a single Transformer model for both intent detection and entity extraction and it produces entities by processing a Transformer's output layer with a CRF layer.

For both CLINC150 and BANKING77, we use the training subset to fine-tune the models. The test subset is used to compute the accuracy metrics. In all our experiments, the models are fine-tuned for 50 epochs using the training set, except for the SpaCy embeddings setup where fine-tuning is performed for 100 epochs.

The results are presented separately, depending on the used classifier. For this set of experiments, we use the following hardware configuration: a system with an Intel(R) Core(TM) i7-9850H CPU @ 2.60GHz processor, 32 GB RAM, and a NVIDIA Quadro T1000 GPU with 4 GB VRAM. The results from Table 1 are obtained without using the Transformer architecture (SklearnIntentClassifier and MitieClassifier). For all the scores in Table 2, the DIETClassifier was used.

**Table 1.** Intent classification accuracy obtained through non-Transformer-based methods. Best performing models for each dataset have their results in bold.

|  | CLINC150 | BANKING77 |
| --- | --- | --- |
| SpaCy Embeddings | **0.8271** | 0.8867 |
| CountVectorsFeaturizer | 0.7418 | **0.9026** |

**Table 2.** Intent classification accuracy obtained by using different language model features extracted from the user input text. Best performing models for each dataset have their results in bold.

|  | CLINC150 | BANKING77 |
|---|---|---|
| BERT Embeddings | 0.8104 | **0.9282** |
| ConveRT Embeddings | **0.8242** | 0.9237 |
| RoBERTa Embeddings | 0.7651 | 0.9192 |
| GPT Embeddings | 0.7956 | 0.9081 |
| GPT-2 Embeddings | 0.7656 | 0.9019 |
| XLNet Embeddings | 0.7627 | 0.9006 |

The accuracy rates obtained by the featurizer based on word counts are slightly lower than those obtained by the featurizers that use neural models pre-trained on English texts (both SpaCy and language model based featurizers).

To better understand classification accuracy and which types of examples are misclassified, we computed the precision, recall, and macro-F1 scores for the language model-based methods. The scores were computed for both individual intent classes and globally for all classes. Table 3 presents the average scores by metric for RASA NLU intent classification. A sample plot of the resulting scores obtained by using the BERT featurizer for the BANKING77 set is presented in Figure A1 in Appendix A.

**Table 3.** Recall, precision, and F1 classification scores obtained using different types of language model featurizers (LMF). Best performing models, in terms of macro-F1 average score, have their results in bold.

| LMF | CLINC150 | | | BANKING77 | | |
|---|---|---|---|---|---|---|
|  | Recall | Precision | Macro-F1 (avg.) | Recall | Precision | Macro-F1 (avg.) |
| BERT | 0.9455 | 0.8236 | 0.8735 | 0.9282 | 0.9317 | **0.9283** |
| ConveRT | 0.9475 | 0.8369 | **0.8816** | 0.9237 | 0.9277 | 0.9241 |
| RoBERTa | 0.9107 | 0.7856 | 0.8349 | 0.9191 | 0.9220 | 0.9192 |
| GPT | 0.9311 | 0.8169 | 0.8614 | 0.9081 | 0.9122 | 0.9084 |
| GPT-2 | 0.9066 | 0.7868 | 0.8332 | 0.9019 | 0.9056 | 0.9018 |
| XLNet | 0.8986 | 0.7813 | 0.8272 | 0.9006 | 0.9052 | 0.9003 |

Similar to the accuracy scores presented in Table 2, the highest F1 scores are obtained by using the ConveRT (for CLINC150) and BERT (for BANKING77) language model featurizers. To check which intents are specifically mistaken for other intents, we plot the confusion matrix of the test set for the BANKING77 dataset when using the ConveRT featurizer (Figure 1). The full confusion matrix is presented in Figure A2 in Appendix B.

The confusion matrix reveals that some of the incorrectly classified examples, i.e., the light-purple hue, denote semantically similar intent labels, which in turn contain semantically similar examples in the training set. In this sense, some examples of similar intents are:

- *card_arrival* vs. *order_physical_card*;
- *pending_top_up* vs. *top_up_reverted*;
- *declined_transfer* vs. *declined_card_payment*;
- *balance_not_updated_after_bank_transfer* vs. *transfer_timing*;
- *virtual_card_not_working* vs. *card_not_working*.

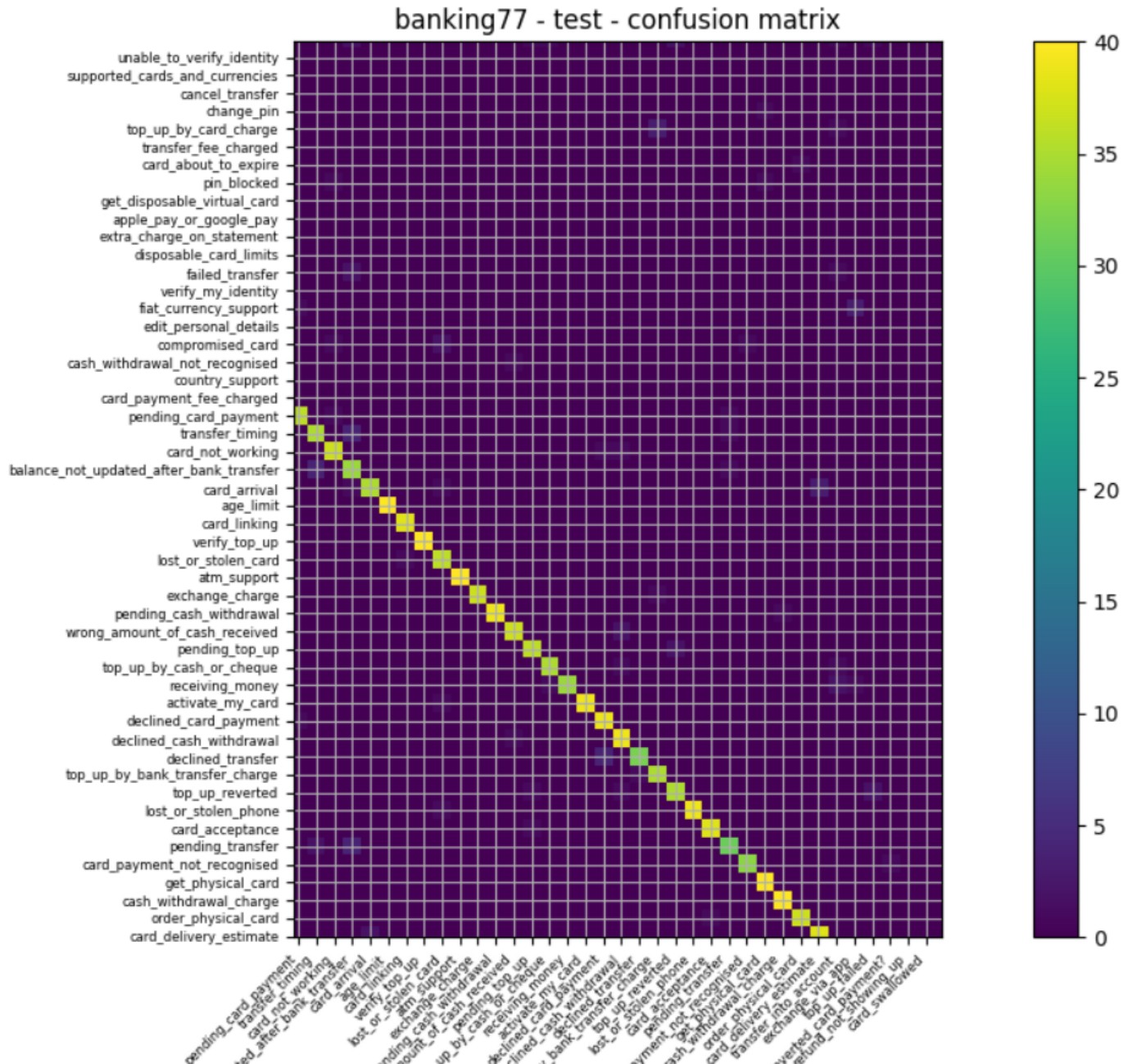

**Figure 1.** Selected section of the confusion matrix obtained for classifying BANKING77 test instances, with a model using the ConveRT language model featurizer. The yellow-green hue represents correctly classified test instances and it represents the main diagonal of the full matrix.

We do not include the confusion matrix for the CLINC150 test set, as the corresponding plot is not easily readable. However, it is plotted and interpreted with the help of a tool that renders it inside a scrollable webpage. Unlike BANKING77, where all test set classes contain exactly 40 instances, for CLINC150, the test set is unbalanced. There are many more *OOS* (out-of-scope) intents compared to the other ones (1000 vs. 30 for each other intent). As a result, most of the misclassifications occur when classifying an *OOS* example.

## 5. Corpus Clustering

As stated before, a standard RASA conversational agent relies on two distinct pipeline stages in order to converse with a user: (1) the NLU component and (2) the dialogue management component. At both levels, the chatbot designer has to provide multiple

learning examples in terms of intents and conversational scenarios and, in order to obtain a robust agent, the examples must be as diverse and numerous as possible. Even though RASA automates to some extent the process of capturing training data through the RASA interactive mode, obtaining an adequate list of examples still remains a tedious and time-consuming task.

On the other hand, there exist many datasets containing conversations that could be used for acquiring the necessary data (e.g., *Cornell Movie-Dialogs Corpus* (https://www.cs.cornell.edu/~cristian/Cornell_Movie-Dialogs_Corpus.html (last accessed on 7 November 2022)) [26], *OpenSubtitles* (https://opus.nlpl.eu/OpenSubtitles-v2018.php (last accessed on 7 November 2022)) [27], etc.). Being able to process them in order to query for similar examples, given a designer's chosen example, would drastically reduce the chatbot design time.

The similarity could be exploited either at *local* level (utterance level) or at *global* level (conversation level), with the latter option being more difficult to tackle. Moreover, standard text similarity metrics (such as cosine distance) could be used to retrieve similar examples. The current impediment is given by the size of each such dataset, which makes a linear search prohibitively slow even for a single example. A possibility for an efficient example retrieval system would rely on pre-computations and searches performed within a subset of the complete dataset. Thus, offloading the intensive computations to a preprocessing step would ensure a smaller retrieval time for a single example.

## 5.1. Method Description

Our method can be regarded as a pipeline which processes raw transcripts, embeds individual sentences in order to obtain dense feature representations, and clusters them in order to shorten similar sentence retrieval time.

### 5.1.1. Data Preprocessing

The proposed system uses the subset of English subtitles from the OpenSubtitles [27] corpus as the training set. The subtitles are encoded as XML files. Each subtitle contains additional markdown data necessary for displaying specific parts of the subtitle at the correct moment. For this set of experiments, we only use the raw text of the subtitles. The timestamps are not necessary and, therefore, are discarded.

The initial set, including time annotations, contains 123 GB of data split across approximately 446,000 subtitle files. The first stage consists of aggregating the text of several files into larger 'record' files to ensure that dataset loading times are minimized. After this step, the dataset's size is reduced to approximately 11 GB of raw subtitles text split across 105 record files, each holding 100 MB of data. The total number of utterances in the resulting corpus is approximately 381 million and each file holds between 3.4 and 3.9 million examples. After manual examination, it was noticed that some movies contain multiple versions of the same transcript, which are most of the time identical. After filtering out duplicate subtitles, the final corpus is reduced to only 140,000 subtitle files, or 4 GB of raw text, split into 37 record files, approximately 100 MB of data each. The number of sentences per record remains unchanged. However, the total number of examples available is lowered to 131 million. The process of creating the record files considers that transcript lines being part of the same movie scene in the initial dataset to not be split across different record files.

### 5.1.2. Embedding, Clustering, and Data Projection

*Local* level similarity can be computed based on either sparse or dense features of the text. Following the success of the Transformer architecture in numerous NLP tasks, several types of embedding vectors obtained through the encoder modules of different Transformer models may be used. Two such options might be:

- A BERT [20] model from Huggingface (*bert-base-uncased*);
- An SBERT (Sentence BERT) [28] model (*all-mpnet-base-v2*).

The BERT model is used by Devlin et al. [20] to demonstrate that for a Named Entity Recognition (NER) task, state-of-the-art accuracy in terms of F1 score is obtained without fully fine-tuning a pre-trained BERT model on the training set at hand. However, the authors extracted contextual embeddings from several hidden layers and used them as input to two BiLSTM layers before applying the final classification layer. The results show that using embedding vectors obtained by concatenating the last four hidden layers produces the best results.

SBERT [28] is based on a pre-trained MPNet language understanding model [29] and fine-tuned on 1 billion pairs of sentences. The objective of pre-training is to predict to which pair a randomly given sentence belongs. In this case, the computed sentence-level embedding vectors have a lower dimensionality of 768 units compared to the solution offered by Devlin et al. [20]. This makes SBERT the preferred alternative when the dataset used is large, as in the case of OpenSubtitles, because precomputed embeddings would require at least four times less storage.

Given an input sentence $x$, retrieving semantically similar instances from a learning set $\mathcal{D}$ of $N$ instances can be achieved by retrieving the sentence $y$, yielding the maximum cosine similarity between the corresponding embedding vectors:

$$y = \underset{t \in \mathcal{D}}{\text{argmax}} \frac{\boldsymbol{emb_x} \cdot \boldsymbol{emb_t}}{\|\boldsymbol{emb_x}\| \cdot \|\boldsymbol{emb_t}\|} \tag{1}$$

From Equation (1), we observe that $\boldsymbol{emb_x}$ is compared against each instance in the dataset $\mathcal{D}$, which becomes unfeasible as the size $N$ of $\mathcal{D}$ increases. In order to speed up the linear search by a constant factor $K$, the current approach proposes to divide all the $N$ learning instances into $K$ disjoint groups. Separation is performed based on embedding vectors $\boldsymbol{emb_t}$ of each sentence in $\mathcal{D}$. For small sizes of $N$, any standard clustering algorithm may be used to aggregate similar embedding vectors into the same cluster [30]. This becomes, however, impractical as $N$ grows, due to large memory requirements in the clustering method. For instance, standard K-means clustering requires to have all data which are to be fitted in memory at once which, in turn, slows the algorithm performance [31]. Considering the dimensions of the gathered dataset, K-means would require $131 \times 10^6 \times 768 \times 4 \approx 375$ GB of memory (4 represents the size in bytes for a standard float).

Instead, *online* clustering algorithms could be used to cluster sentences. One such example is *mini-batch K-means* [32]. Similar to standard K-means, it optimizes the same non-convex objective function, while iteratively processing batches of the input data $\boldsymbol{X}$. Equation (2) presents the *mini-batch K-means* optimization function, where $\boldsymbol{X}$ contains the embeddings of the instances in $\mathcal{D}$ and $\boldsymbol{c_t}$ represents the embedding of the centroid of the cluster where $\boldsymbol{t}$ is assigned.

$$\mathcal{L} = \sum_{t \in \boldsymbol{X}} \|\boldsymbol{t} - \boldsymbol{c_t}\|^2 \tag{2}$$

Even though processing data in batches allows to construct and process of large amounts of embedding vectors, in a streaming manner, it might also have the disadvantage of possibly invalidating previous cluster assignments, e.g., $t$ assigned to cluster $c_t$ at timestep $T$ might need to be reassigned to a different cluster after processing the next batch at timestep $T + 1$ since $c_t$ might suffer significant modifications. However, depending on the sampled subset of instances, $t$ might incorrectly remain assigned to the same cluster $c_t$. In practice, both standard K-means and mini-batch K-means converge to similar cluster assignments. During the cluster center update step, mini-batch K-means attempts to move the cluster centers as little as possible away from the previous cluster centers, by considering them as well in the update equation.

In the current implementation, fitting the data through mini-batch K-means is done for a fixed number of steps rather than until a given convergence criterion is met. After fitting the current examples, the embedding vectors and corresponding cluster labels are stored on the disk, to allow the processing of the next batch of embedding vectors. Furthermore, the fitted K-means object is also stored. Fitting the next batch of data must consider previously

fitted data and the K-means object must also be persisted for later usage in the inference phase.

Having the instances from the learning set embedded and clustered, performing inference for a given test instance $q$ consists of the steps described in Algorithm 1:

- Obtain the embedding vector $\boldsymbol{emb_q}$ for $q$ based on the embedder $E$ used for performing clustering (Line 1).
- Identify the closest cluster center $c_q$ (bin) to $\boldsymbol{emb_q}$, as computed through mini-batch K-means, i.e., compute the cosine similarity against all $K$ clusters and return the most similar cluster center (bin) (Line 2).
- Identify the closest embedding vector to $\boldsymbol{emb_q}$, in the current cluster $c_q$ (bin), i.e., compute the cosine similarity against all learning examples assigned to cluster $c_q$ (bin), and return the most similar example (Line 3). Note that the comparison is performed only within a restricted number of subsets of the initial data as we assume roughly uniform splitting of the initial $N$ examples across the $K$ clusters. Thus, the computations involve only $\frac{N}{K}$ cosine similarity computations.
- Lookup in the original corpus and retrieve $p$ the natural language sentence paired with the index based on the embedding vector index (Lines 4 and 5).

---

**Algorithm 1** Inference steps for a given test example $q$

---

**Input:** $E$ embedder, $q$ inference sentence
**Output:** $p$ the <sentence, index> pair

1: $\boldsymbol{emb_q} \leftarrow E(q)$                ▷ Obtain the embedding vector
2: $c_q \leftarrow$ closest cluster to $\boldsymbol{emb_q}$          ▷ Identify the closest cluster
3: $m_q \leftarrow$ closest example to $\boldsymbol{emb_q}$ in $c_q$
4: $p \leftarrow index(m_q)$                ▷ Index in the original corpus
5: **Return** $p$

---

Due to the online nature of the clustering algorithm and to counteract the possibility of early learning instances being assigned to a wrong bin, the second and third steps above can check more than one bin and example. This idea is inspired by *Beam Search*, a greedy decoding algorithm used in other NLP tasks (e.g., dialogue generation, machine translation, etc.), where multiple candidates in an implicit graph structure are explored in a breadth-first search manner. This might prove useful for detecting embedding vectors falling under a very similar bin, which might rank just below the closest bin in terms of cosine similarity to the inference embedding vector.

It is certain that not all of the sentences retrieved through the method described above would positively impact the quality of the corpus which is to be augmented. However, given a limited set of hand-chosen learning examples, one can train a *weak* classifier with the initial set of examples. The additional examples retrieved by the system can be filtered based on the classifier class output probabilities, i.e., if an example is assigned to a specific class with a probability greater than a threshold, then the example will be further considered for augmenting the corpus; otherwise, the low probability will lead to the dismissal of the example.

This initial technique of example filtering might not drastically improve classification accuracy, i.e., for high values of the confidence threshold (e.g., 0.9), the model might choose only examples which do not bring any additional information. A different approach would be to filter examples according to a semi-supervised approach, treating the initial learning set as labeled, and the set of retrieved examples as unlabeled. This idea is inspired by FixMatch, a method initially developed and applied for computer vision tasks [33].

While storing precomputed sentence embedding vectors decreases the lookup time, storage requirements are particularly high; i.e, the entire set of 131 million, 768-dimensional vectors requires approximately 390 GB of storage space. In order to reduce the amount of storage needed, dimensionality reduction algorithms can be used in order to downsample the embedding vectors. A different approach could be to use a different embedder, which

produces lower dimensional embeddings. However, the main issue remains, as we do not know any embedder that might output vectors of only 32 or 64 dimensions, so we do not follow this path of experiments.

Similar to the clustering algorithm limitations, one requirement for the dimensionality reduction algorithm is that it must be able to process the data iteratively since they would not fit into the memory all at once. As a result, IPCA (Incremental PCA) is successfully used to downsample the set of vectors to 64 and 32 dimensions, while preserving semantic similarity [34]. The resulting sets take only 76 GB and 43 GB of storage, respectively.

*5.2. Few-Shot Learning*

A natural use case of this similar example retrieval system is the few-shot training scenario. Such a setup examines the performance of a model trained with a limited, small number of examples. Models may either make use of transfer learning or corpus augmentation techniques in order to increase the desired performance metric.

Considering a learning set $\mathcal{D}$ with $C$ classes, each class has $n_C$ examples. The augmentation process consists of retrieving $K \times L$ additional examples for each of the $n_C$ examples, with $K$ being the number of clusters to check and $L$ being the 'beam' size in each cluster. The retrieved examples may be subject to further filtering or post-processing step, in order to minimize the noise introduced in the dataset. In our experiments, we test two different methods for filtering:

(1) Use a model trained on the initial data to classify the retrieved examples. The examples classified with confidence exceeding a fixed threshold (i.e., 0.8 or 0.9) are kept, while the others are discarded.

(2) Remove stop words from both the initial examples and the retrieved examples. Then, compute the set intersection over the tokens of a candidate sentence and the complete set of tokens of the initial examples. Only examples producing an intersection size over a given threshold (i.e., 1 or 2) are kept. Intuitively, this method forces to some extent the retrieved examples to be lexically similar to the initial examples.

## 6. Evaluation and Results

To evaluate the proposed similar sentence retrieval system, we use it to augment the intent classification training sets and, then, we evaluated the models trained with the augmented data on the unmodified test sets, using RASA's DIETClassifier. The first scenario aimed to verify to what extent does the number of clusters impact the classification accuracy. Thus, we experiment with 512 and 1024 clusters. In both setups, the complete training sets of BANKING77 and CLINC150 are used to train initial DIETClassifier models. These classifiers are subsequently used for classifying the additionally extracted examples. Only examples classified with at least 0.9 confidence are used for augmenting the training sets. For this set of experiments, we use the same hardware configuration as for the intent classification experiments (Section 4.2). For result reproducibility, we will make the code publicly available on GitHub, in the following repository: https://github.com/Gabriel-Bercaru/CorpusClustering.

For each example in the initial training sets, the top $K = 1$ cluster is inspected, retrieving the $L = 1$ similar example. In each setup, the initial training set sizes were 10,080 examples for BANKING77 and 15,251 examples for CLINC150. For augmentation, in the first phase 10,080 and 15,251 examples are retrieved. Out of these, only 3% and, respectively, 20% of them are classified with a confidence of at least 0.9 (294 for BANKING77 and 3043 for CLINC150), resulting in augmented set sizes of 10,374 and 18,294 examples. For each setup, 30 DIETClassifier models with different randomly initialized parameters are trained on the initial and augmented datasets. We measure the mean accuracy and the standard deviation for each setup (Table 4). We should note that for each setup, the BERT featurizer is kept fixed throughout all the experiments. Figure 2 presents the clustering results.

**Table 4.** Mean accuracy and standard deviation for the first augmentation method, in which a pretrained classifier is used for classifying additional examples. Bold text denotes the best performing model.

|  | **CLINC150** | **BANKING77** |
|---|---|---|
| Original data | $0.8037 \pm 0.0039$ | $0.9305 \pm 0.0031$ |
| Augmented data—1024 clusters | $0.8036 \pm 0.0046$ | $0.9296 \pm 0.0027$ |
| Augmented data—512 clusters | $\mathbf{0.8058 \pm 0.0052}$ | $\mathbf{0.9305 \pm 0.0022}$ |

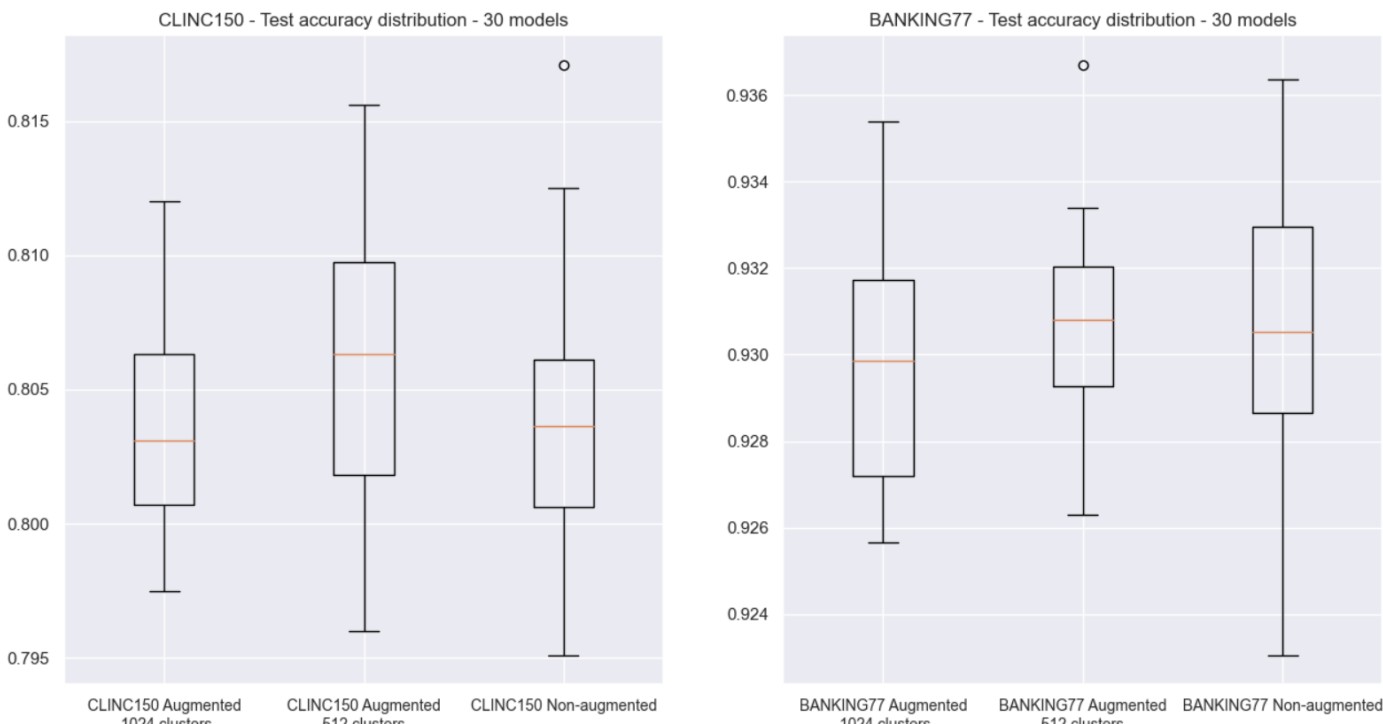

**Figure 2.** Box and whisker plots obtained when training an ensemble of 30 models for each combination of dataset and augmentation method. Augmented training sets do not improve the mean accuracy on the test set, but reduce variance across models. Whiskers extend from the lower to the upper quartile of the data.

As expected, the results are approximately the same because, firstly, the retrieved examples are selected to be as similar as possible to the ones already in the dataset and secondly, at least in the case of BANKING77, the number of retrieved examples is rather small. As already mentioned, a semi-supervised approach might help in future research to extract more meaningful examples out of the 'unlabeled' automatically retrieved set.

The second set of experiments is conducted to evaluate the sentence retrieval system in a few-shot scenario. In this setup, the training sets of BANKING77 and CLINC150 are sequentially restricted to only $k \in \{2, 3, 5, 10\}$ examples per class. The corpus clustering method is then used to artificially increase the number of examples available, based on the initial, limited number of examples. For each initial example $x_i$, a similar sentence $y_i$ is retrieved. In the end, all retrieved $y_i$ are aggregated and combined with the initial learning set and the duplicates are removed.

To minimize the number of noisy examples which are added to the learning set, the following heuristic is tested. When attempting to add a retrieved candidate $y_i$ to a class $C$, first compute its set of unique tokens. Stop words are removed before set computation. Next, perform a set intersection with the set of tokens corresponding to all initial examples

$x_i$ in the class $C$. Only add the example if the set intersection size exceeds a given threshold $t \in \{0, 1, 2\}$. This heuristic attempts to include only examples which are somewhat similar to the initial ones. During testing without the heuristic filtering, we observed that some unrelated examples are added to the learning set and, thus, we introduce this heuristic to avoid this issue. Evaluation of the few-shot setups is performed by training an ensemble of 10 different DIETClassifier models in each configuration. Table 5 presents the mean accuracy and its standard deviation.

**Table 5.** Mean accuracy and standard deviation obtained for the corpus augmentation method in the few-shot scenario. In each augmentation setup, $t$ denotes the stop word (SW) filtering threshold. Note: bold marks the model with the highest mean accuracy.

| Few-Shot-Scenario | Filtering | CLINC150 | BANKING77 |
|---|---|---|---|
| k = 2 | no augmentation | $0.2127 \pm 0.0150$ | $0.1983 \pm 0.0205$ |
| | augmentation, t = 0 | $\mathbf{0.3734 \pm 0.0087}$ | $\mathbf{0.3336 \pm 0.0169}$ |
| | augmentation, t = 1 | $0.2302 \pm 0.0107$ | $0.2618 \pm 0.0216$ |
| | augmentation, t = 2 | $0.2347 \pm 0.0123$ | $0.2379 \pm 0.0162$ |
| k = 3 | no augmentation | $0.3941 \pm 0.0098$ | $0.3875 \pm 0.0155$ |
| | augmentation, t = 0 | $\mathbf{0.4793 \pm 0.0128}$ | $\mathbf{0.4562 \pm 0.0157}$ |
| | augmentation, t = 1 | $0.4575 \pm 0.0106$ | $0.4371 \pm 0.0115$ |
| | augmentation, t = 2 | $0.3850 \pm 0.0236$ | $0.4204 \pm 0.0159$ |
| k = 5 | no augmentation | $0.5273 \pm 0.0144$ | $0.6007 \pm 0.0145$ |
| | augmentation, t = 0 | $\mathbf{0.5617 \pm 0.0088}$ | $\mathbf{0.6199 \pm 0.0130}$ |
| | augmentation, t = 1 | $0.5572 \pm 0.0088$ | $0.6140 \pm 0.0117$ |
| | augmentation, t = 2 | $0.5491 \pm 0.0094$ | $0.6028 \pm 0.0158$ |
| k = 10 | no augmentation | $0.6622 \pm 0.0125$ | $0.7667 \pm 0.0074$ |
| | augmentation, t = 0 | $0.6570 \pm 0.0063$ | $0.7544 \pm 0.0062$ |
| | augmentation, t = 1 | $0.6657 \pm 0.0077$ | $0.7648 \pm 0.0059$ |
| | augmentation, t = 2 | $\mathbf{0.6681 \pm 0.0058}$ | $\mathbf{0.7669 \pm 0.0086}$ |

## 7. Discussion

Regarding the first set of experiments, in which we test different featurizers, it can be observed that embeddings provided by Transformer neural models help improve the intent classification accuracy, with BERT and ConveRT embeddings performing the best for both BANKING77 and CLINC150 datasets.

For the corpus clustering part, two sets of experiments are conducted. The first one consists in analyzing whether augmenting the training sets of BANKING77 and CLINC150 helps improve intent classification accuracy. As the results in Table 4 show, the method brings minor improvements in terms of classification mean accuracy, also with a reduction in variance. Moreover, the number of clusters used for grouping together similar examples seems to bring little influence, as in both cases, the classification means accuracy is approximately equal, with a small improvement when using 512 clusters.

For the second set of experiments regarding the few-shot scenario, we restrict the training sets of BANKING77 and CLINC150 to only 2, 3, 5, or 10 examples per intent. The corpus clustering method is then used to artificially increase the training set sizes. In the best-case scenario, the sizes are doubled. However, in most cases, duplicate similar examples are retrieved and, therefore, they are removed. Moreover, additional examples are removed according to the heuristic described in Section 6. In this setup, we observe that the proposed corpus clustering method helps improve the classification accuracy, in the best case leading to 16% accuracy increase for CLINC150 when $k = 2$ (Table 5) and a 14% accuracy increase for BANKING77 when $k = 2$ (Table 5). As more of the original training examples become available, the proposed method still increases the mean classification accuracy, but to a smaller extent. Including additional original training data will most likely result in even smaller improvements and will ultimately produce results similar to those presented in Table 4.

When interpreting the results, one should consider that they reflect the scores obtained when clustering based on reduced versions of the sentence embedding vectors, i.e., 32 dimensions, are used. The used sentence embedder (SBERT) produces 768-dimensional vectors. We hypothesize that using the full embedding vectors, with no dimensionality reduction applied, would lead to the retrieval of more meaningful examples, increasing the reported accuracy values. However, the retrieval time per example increases as well. In the 32-dimensional embedding vectors setup, the retrieval time per example is approximately 0.2–0.3 s, while for the 768-dimensional embeddings, the retrieval time is approximately 0.6 s. The exploration of this hypothesis is left as part of a future investigation.

## 8. Conclusions

In this work, we examine the problem of intent classification as part of a conversational agent pipeline. First, we discuss how existing systems perform in terms of intent classification—answering ($Q_1$) and achieving objective ($O_1$). Then, we define a method for clustering large corpora, to efficiently retrieve examples that are similar to a user-supplied query. Our method consists of several preprocessing stages, such as embedding movie transcripts, online clustering, and data projection. By making use of precomputations and data partitioning into clusters, we achieve low inference duration—answering ($Q_2$) and achieving objective ($O_2$). We automatically process 123 GB of raw movie subtitles data, available as part of the OpenSubtitles dataset—answering ($Q_3$) and achieving objective ($O_3$). The corpus clustering method is shown to bring minor improvements in terms of classification accuracy when the full training sets are available. Moreover, we also examine to what extent the method helps improve the accuracy when a limited number of examples are available. Our results have shown that the intent classification accuracy is raised by up to 16%, in the most favorable case, where only two labeled examples per class are available. Our proposed method achieves retrieval times as low as 0.2–0.3 s per example and is shown to bring statistically relevant improvements in intent classification scenarios in which training data are scarce—answering ($Q_4$) and achieving objective ($O_4$).

For tasks in which large datasets are available, our method does not introduce significant improvements; this is due to the fact that large datasets expose a high degree of example diversity and additional retrieved examples might not bring in additional useful information. However, for small datasets, our method helps improve the diversity of the examples, leading to larger accuracy scores, as shown by our research.

In future work, we plan to expand the corpus clustering method in order to further reduce the retrieval time per example. One such possibility would be to move to a hierarchical clustering approach. During the experiments, it was observed that some clusters are considerably larger than others; the loading time for such clusters becomes a bottleneck. A solution would be to identify the large clusters and further group their elements into smaller sub-clusters, in order to minimize the cluster loading time during the example retrieval phase.

Another possible direction that we will investigate is to use a semi-supervised learning approach in order to filter retrieved examples. In this work, we investigated the effect of filtering all the retrieved examples based on a pre-trained classifier confidence threshold. However, as the results show, this does not lead to major improvements in classification accuracy. Using a semi-supervised approach, in which the full set of retrieved examples is regarded as unlabeled, would possibly lead to better choices when filtering the examples, yielding more meaningful augmentations.

As a possible future application, we plan to evaluate how our proposed pipeline performs in augmenting real-world conversational scenarios. We plan to implement a conversational agent focused on the interaction during interviews. Its learning set is an ideal candidate for evaluating our data augmentation method. Since our method mainly deals with dataset augmentation, there is currently no plan to use it in a real-time scenario.

**Author Contributions:** Conceptualization, G.B., C.-O.T., C.-G.C. and T.R.; methodology, G.B., C.-O.T., C.-G.C. and T.R.; software, G.B.; validation, G.B., C.-O.T., C.-G.C. and T.R.; formal analysis, G.B., C.-O.T., C.-G.C. and T.R.; investigation, G.B., C.-O.T., C.-G.C. and T.R.; resources, G.B., C.-O.T., C.-G.C. and T.R.; data curation, G.B.; writing—original draft preparation, G.B., C.-O.T., C.-G.C. and T.R.; writing—review and editing, G.B., C.-O.T., C.-G.C. and T.R.; visualization, G.B.; supervision, C.-O.T., C.-G.C. and T.R.; project administration, C.-G.C. and T.R.; funding acquisition, T.R. All authors have read and agreed to the published version of the manuscript.

**Funding:** This work was funded by the Romanian Ministry of European Investments and Projects through the Competitiveness Operational Program (POC) project "HOLOTRAIN" (grant no. 29/221_ap2/07.04.2020, SMIS code: 129077).

**Institutional Review Board Statement:** Not applicable.

**Informed Consent Statement:** Not applicable.

**Data Availability Statement:** The CLINC150 and BANKING77 datasets used in this study are publicly available at https://github.com/clinc/nlu-datasets (last accessed 15 December 2022). The OpenSubtitles dataset is publicly available at https://opus.nlpl.eu/OpenSubtitles-v2018.php (last accessed 15 December 2022).

**Conflicts of Interest:** The authors declare no conflicts of interest.

## Abbreviations

The following abbreviations are used in this manuscript:

| | |
|---|---|
| BERT | Bidirectional Encoder Representations from Transformers |
| CRF | Conditional Random Field |
| DIET | Dual Intent-Entity Transformer |
| GPT | Generative Pre-Training |
| ID | In-Domain |
| LMF | Language Model Featurizer |
| NER | Named Entity Recognition |
| NLP | Natural Language Processing |
| NLU | Natural Language Understanding |
| OOD | Out-of-Domain |
| OOS | Out-of-Scope |
| RoBERTa | Robustly Optimized BERT pretraining Approach |
| SBERT | Sentence BERT |
| SVM | Support Vector Machines |
| SW | Stop Word |
| TED | Transformer Embedding Dialogue |
| TF-IDF | Term Frequency-Inverse Document Frequency |

## Appendix A. Intent Classification: Recall, Precision, and F1 Scores

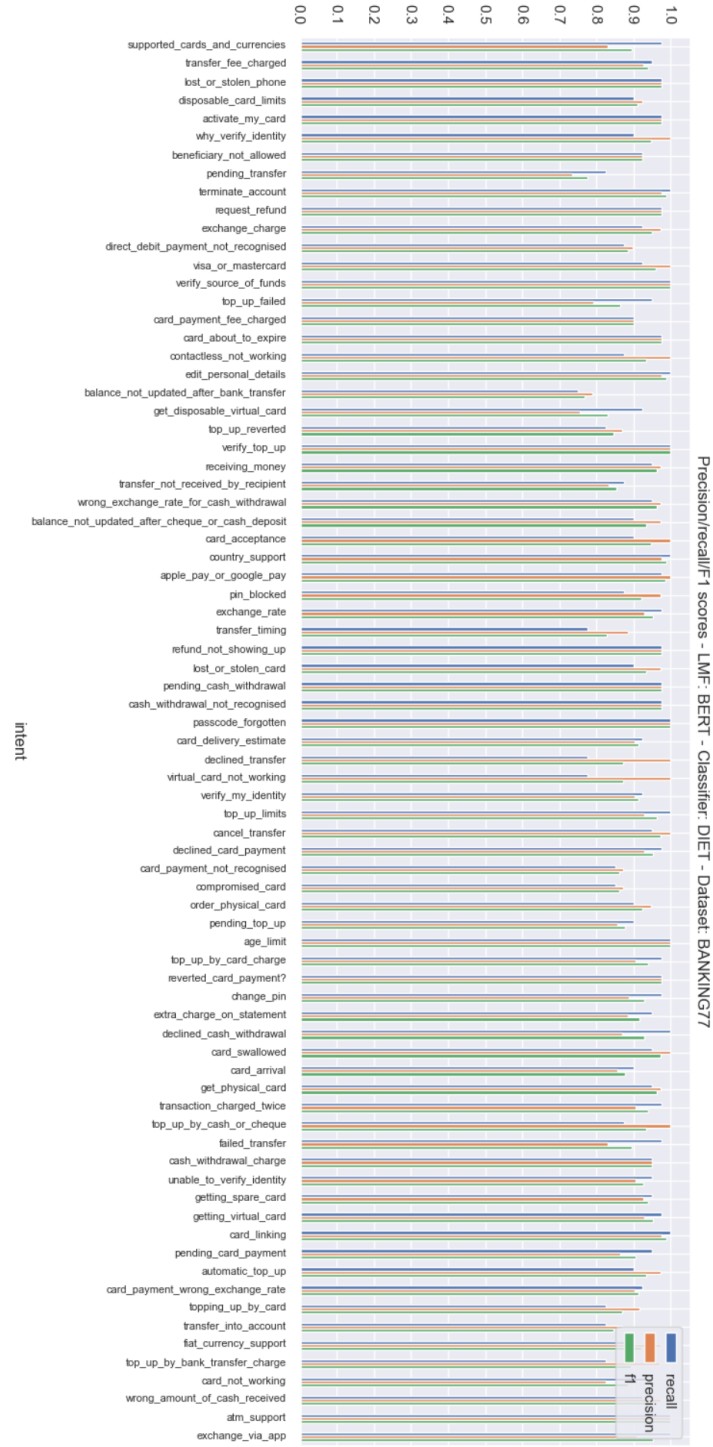

**Figure A1.** Recall, precision, and F1 scores obtained for intent classification on the BANKING77 test set using the BERT model as a language featurizer.

**Appendix B. Intent Classification: Confusion Matrix**

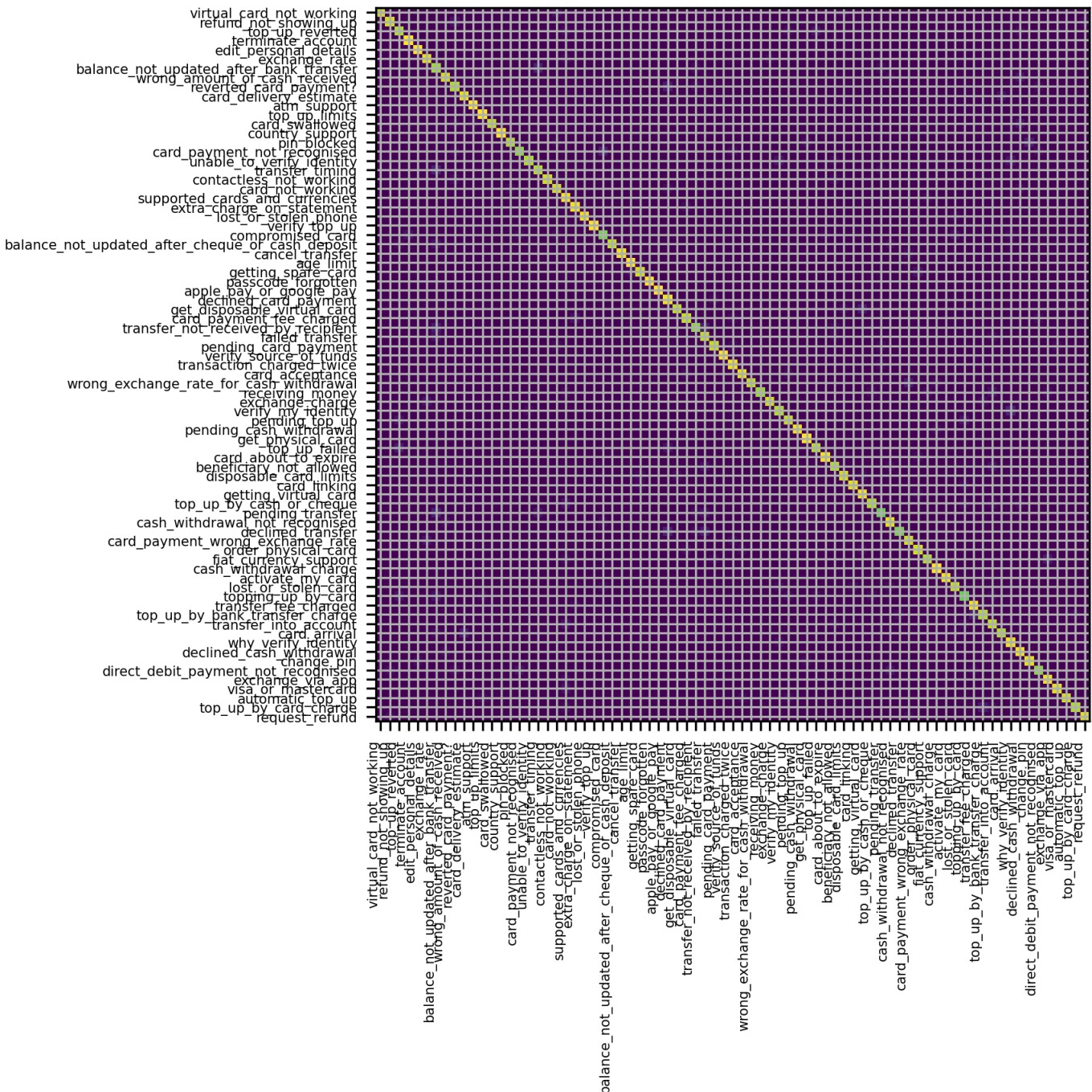

**Figure A2.** Full confusion matrix obtained for classifying BANKING77 test instances, with a model using the ConveRT language model featurizer. The yellow-green hue represents correctly classified test instances and it represents the main diagonal of the full matrix.

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
