# Peer review of "Improving Intent Classification Using Unlabeled Data from Large Corpora"

_mathematics, doi:10.3390/math11030769_

Round 1
Reviewer 1 Report
The study concerns the problem of intent classification. The authors offer their method for efficiently augmenting textual corpora using large datasets. The paper is well written and contains all required sections.
Some comments:
1) Abbreviation NLP is not defined in the abstract
2) The use of footnotes in the text is questionable, references to the bibliography will be more acceptable here.
3) The description of datasets in Section 2 is repeated in Section 4. It would be reasonable to leave only one
4) The text at the bottom of the Figure 1 is not readable, the same for the Figure A2.
5) References to questions and objectives from the introduction make it difficult to read the conclusion
Reviewer 2 Report
The authors have presented an interesting work. But the quality of the manuscript can be improved by addressing below points:
- The authors have not mentioned the problem statement in a separate section. The problem statement section should be included in the manuscript.
- How the present work is different from the previous work? What is the novelty of the present work? These things should be addressed.
- What is the mathematical mechanism behind the RASA framework?
- What is the future scope of this work? How can be it implemented in a real-time scenario?
Reviewer 3 Report
The paper utilizes unlabeled data from large corpora to improve Intent Classification. Tests were performed on real-world datasets, the authors said that their methods enable an increase of text classification accuracy and are important for any NLP or NLU task in which labeled training data is scarce or expensive to obtain.
The paper is interesting, easy to follow, and the plagiarism report says the uniqueness of the text is 91% (Turnitin).
I have some comments are as follows:
1- Should present related work in tabulated form would be desirable to perform meta-analysis of available work and establish ground for proposed work.
2- Too many sections breaking the continuity of the manuscript. Please revise it.
3- The research questions are not clearly answered.
4- Further introductions to DIETClassifier are needed.
5- 2 -> two
6- Conclusions: The paper lacks a conclusion but rather a discussion which is a mere summary of the paper. The authors are expected to explain the reason why the results appear like that more formally.
7- At least 3 citations from Mathematics should be added.
